# K_0.5_Na_0.5_NbO_3_-SrTiO_3_/PVDF Polymer Composite Film with Low Remnant Polarization and High Discharge Energy Storage Density

**DOI:** 10.3390/polym11020310

**Published:** 2019-02-12

**Authors:** Chuntian Chen, Lei Wang, Xinmei Liu, Wenlong Yang, Jiaqi Lin, Gaoru Chen, Xinrui Yang

**Affiliations:** 1School of Science, Harbin University of Science and Technology, Harbin 150080, China; chenchuntian730@sohu.com (C.C.); 15904613620@163.com (L.W.); linjiaqi@hrbust.edu.cn (J.L.); yxinruihit@163.com (X.Y.); 2Laboratory of Engineering Dielectrics and Its Application, Ministry of Education, Harbin University of Science and Technology, Harbin 150080, China; 3State Grid Fuzhou Electric Power Supply Company, Fuzhou 350009, China; chengaoru2009@163.com

**Keywords:** recoverable energy density, remnant polarization, SrTiO_3_, PVDF, polymer composites

## Abstract

A high recoverable energy storage density polymer composite film has been designed in which the ferroelectric-paraelectric 0.85 (K_0.5_Na_0.5_NbO_3_)-0.15SrTiO_3_ (abbreviated as KNN-ST) solid solution particles were introduced into polyvinylidene fluoride (PVDF) polymer as functional fillers. The effects of the polarization properties of K_0.5_Na_0.5_NbO_3_ (KNN) and KNN-ST particles on the energy storage performances of KNN-ST/PVDF film were systemically studied. And the introduction of SrTiO_3_ (ST) was effective in reducing the remnant polarization of the particles, improving the dielectric properties and recoverable energy storage density of the KNN-ST/PVDF films. Compared to KNN/PVDF films, the dielectric permittivity of composite films was enhanced from 17 to 38 upon the introduction of ST. A recoverable energy storage density of 1.34 J/cm^3^ was achieved, which is 202.60% larger than that of the KNN/PVDF composite films. The interface between the particles and the polymer matrix was considered to the enhanced dielectric permittivity of the films. And the reduced remnant polarization of the composites was regarded as the improving high recoverable energy storage density. The results demonstrated that combing ferroelectric- paraelectric particles with polymers might be a key method for composites with excellent dielectric permittivity, high energy storage density, and energy efficiency.

## 1. Introduction

Polymeric composites with high energy density have been attracting increasing attention in recent years, which possess both the outstanding dielectric properties of selected functional inorganic fillers and the high breakdown strength, machinability, and mechanical performance of the polymer matrix. It is regarded as the potential candidates in the field of energy storage owing to their excellent mechanical and dielectric properties [1,2,3,4,5,6]. The energy storage density (W1) can be calculated by using the following relation.
(1)$$W_{1}=\int EdD,$$
where E is the applied electric filed, and D is the electric displacement.

The energy storage efficiency (η) can be presented as Equation (2).
(2)$$\eta =W_{1}/W_{1}+W_{2},$$
where W_1_ is recoverable energy storage density, and W_2_ is energy loss density. According to Equation (1), the high saturated polarization and the low remnant polarization can indicate the recoverable energy storage density of the composites.

Polymer material has been widely used in the field of energy storage. Although polymer material has good mechanical properties and high breakdown electric field, it has lower dielectric constant. In order to improve the dielectric constant, an effective strategy that combining the polymer with the functional inorganic fillers has been carried out, and the electrical, optical or magnetic performance can be tailored by adjusting the shape and size or modified surface of the filled particles [7,8,9,10]. Recently, the composition of inorganic fillers into polymer matrix has been reported to enhance the energy storage density of polymer composites. Puli, V.S. et al. reported that the polymer grafted BZT-BCT composite thick film have shown an improved dielectric constant of 56, and compare to the pure PMMA film, the energy storage density of BZT-BCT/PMMA composite film has a significant improvement from 10.3 J/cm^3^ to 22.5 J/cm^3^ [11]. A maximal dielectric constant was 40.8 for the PVDF polymer composite films with 50 vol% PZT particles at 1 kHz, which was 4.77 times higher than that of the pristine PVDF film. And the recoverable energy density 6.41 J/cm^3^ was achieved in 10 vol% PZT/PVDF composite films [12]. Hu et al. investigated that high dielectric constant and high energy density are obtained in PVDF composite contained with core-shell structure TiO_2_@BaTiO_3_ nanoparticles simultaneously. The dielectric constant was improved from 9.2 for pure PVDF to 19.6 for the 10 vol% nanocomposite. The maximal discharge density of 8.78 J/cm^3^, which is 57% larger than that of pure PVDF [13]. And a novel three-layer structure KTa_0.5_Nb_0.5_O_3_/PI nanocomposite film with high discharge energy density was reported by Chen et al. The maximal discharge density of 3.0 J/cm^3^ was obtained for nanocomposite film, which is comparatively higher than pure PI film [14]. Previous work has successfully increased recoverable energy storage density, however, the low energy efficiency and high losses are still the critical restrictions for composites. This problem can be ascribed to two reasons: one is the interfacial polarization caused by the significant difference in dielectric constant and conductivity between the polymer matrix and the fillers [15]. And the other one is that the high dielectric constant fillers usually show a high remnant polarization [16,17,18]. Compare to ferroelectric materials, SrTiO_3_ with cubic phase structure possess a relatively high dielectric constant under ambient environment [19,20,21]. The excellent breakdown strength and the liner dielectric property of ST system account for its low energy dissipation and suitability for energy storage applications [22,23].

Hence, in this paper, 0.85(K_0.5_Na_0.5_NbO_3_)-0.15TiSrO_3_/PVDF polymer composite films were designed and prepared, and the phase, microstructure, dielectric properties and energy storage density of that were investigated and discussed in detail. The remnant polarization of the composite films reduced 33.33% upon the ST particles. A significant improved recoverable energy storage density of 1.34 J/cm^3^ was obtained in the PVDF composite film with 12 vol% KNN-ST, and the maximal energy storage efficiency of 74.68% was obtained in 3 vol% KNN-ST/PVDF composite films.

## 2. Materials and Methods

### 2.1. KNN-ST Fillers Preparation

The 0.85(K_0.5_Na_0.5_NbO_3_)-0.15SrTiO_3_ (abbreviated as KNN-ST) particles were synthesized via conventional solid-state reaction route. The K_2_CO_3_ (99.0%), Na_2_CO_3_ (99.8%), Nb_2_O_5_ (99.99%), SrTiO_3_ (99.8%) and TiO_2_ (99.99%) were dried and weighed according to stoichiometric ratio as initial powders. These precursor powders were ball mixed 12 h by using zirconia balls for well mixture with ethyl alcohol (the quality ratio is 1:1) as grinding media. And the dried powders were sintered at 950 °C for 4 h. Finally, the sintered powders were ball mixed another 12 h and dried, which was used as inorganic fillers for preparing composite films. The KNN powders were also prepared by solid state reaction method. The sintering process was carried at 850 °C for 4 h, which is the only difference with KNN-ST particles.

### 2.2. KNN-ST/PVDF Polymer Composites Films Preparation

The sol-gel method was taken to prepare the KNN/PVDF and KNN-ST/PVDF polymer composites films (with 3 vol%, 6 vol%, 9 vol% and 12 vol% loading concentration, respectively). The KNN and KNN-ST powders and PVDF powders were used as raw materials, respectively. And the dimethylformamide (DMF) were used as the solvent. Firstly, the powders were suspended into DMF with 30 min ultrasonic concussion. Then, the stoichiometric PVDF were added into the solution with stirring for 3 h. In order to prevent the residual bubbles in the solutions, the precursor solutions were placed into a vacuum drying oven for 2 h. And then, the precursor solutions were cast on glass plates by an automatic film applicator. The obtained polymer composite films were heated at 100 °C for 24 h to evaporate the solution. Finally, press the composite films with a vulcanizing press. The thickness of the films ranges from 20 to 40 μm. And the detailed preparation process was shown in Figure 1.

### 2.3. Characterization

The crystal structures of the KNN and KNN-ST fillers were characterized by an X-ray diffractometer (XRD; philips X-pert pro Diffractometer, Almelo, The Netherlands) with Cu Kα radiation. The surface microstructure of fillers and polymer composites films were characterized by scanning electron microscopy (SEM; Quanta 200FEG, FEI, Hillsborough, OR, USA). In order to investigate the dielectric properties of the composite films, the silver electrodes were sputtered on both sides of films, and the dielectric properties were tested by a precise impedance analyzer (Agilent 4249A, Palo Alto, CA, USA) ranges from 100 Hz to 10 MHz. The electric breakdown strength of composite films was investigated by a dielectric withstand voltage test (YD2013, Changzhou Yangzi Electric Co, Ltd., Changzhou, China).

## 3. Results

An X-ray diffraction testing was taken to study the crystal structures of KNN and KNN-ST particles. The XRD patterns of two samples were shown in Figure 2. The result reveals that both the KNN and KNN-ST particles present a single perovskite phase. The orthorhombic phase of KNN particles were characterized by (202) and (020) peak around 45° [24]. And the phase structure of the KNN-ST particles reveals a pseudo-cubic structure. Compared to KNN fillers, no split peak could be found for KNN-ST particles. The result verifies that SrTiO_3_ could be introduced into the KNN cell, and a new compound has been formed.

In order to investigate the microstructure of KNN and KNN-ST particles, a scanning electron microscopy was performed at room temperature, as shown in Figure 3a,b. The average grain size of pure KNN and KNN-ST fillers were about 2 μm and 350 nm, respectively. It can be proved that SrTiO_3_ can remarkably inhibit the growth of grains. And the grain sizes of KNN and KNN-ST grown in the high temperature process. Compared with KNN, the crystal structure of KNN-ST was disordered by the introduction of ST. The disordered crystal structure and the growth energy restricted the grain growth of KNN-ST particles. The decrease of particle size has a great role in promoting the breakdown strength and discharge storage density.

The dispersion state of fillers in matrix is an important factor in determining the dielectric properties of the composite films. As shown in Figure 3c–m, both KNN and KNN-ST particles are well-dispersed in the PVDF matrix for all samples. With the increase of loading content, the agglomeration could be found. Besides, there are potholes appear in KNN-ST/PVDF composited films with high loading concentration, it could be attributed to that there are some bubbles run over between the film and the glass, which may lead to the decline of mechanical properties and breakdown strength for the composite films [25].

The dielectric performances of KNN/PVDF and KNN-ST/PVDF composite films with different volume fraction were measured. And the frequency ranged from 100 Hz to 1 MHz at room temperature. Figure 4 shows the dielectric constant and dielectric loss of pure PVDF and KNN/PVDF composite films, respectively. And Figure 5 shows the dielectric constant and dielectric loss of KNN-ST/PVDF composite films. The results reveal that the dielectric constant of composites films gradually increase with the increasing of fillers, which further indicates the dielectric constant of composites films can be tailored by changing the volume fraction of fillers [26,27,28]. And the dielectric constant is about 38 at 100 Hz in the composites with a loading of 12 vol% KNN-ST, which increased 140% than that of the 12 vol% KNN/PVDF composite films. This remarkable enhancement in dielectric constant should be attributed to the higher dielectric constant of the ST and the Maxwell-Wagner-Sillars (MWS) interfacial polarization, which is mainly caused by the large difference in the dielectric constant and conductivity between the fillers and the polymer matrix [29]. With the increase of frequency, the dielectric constant of KNN/PVDF and KNN-ST/PVDF composite films decrease gradually. This can be attributed to the fact that a part of polarizations can’t achieve the response to frequency.

Under the lower frequency, the dielectric loss caused by interfacial polarization plays a leading role for composite films. Hence, the dielectric loss of KNN/PVDF and KNN-ST/PVDF composite films increased with the increasing of loading content. It is worth noting that the dielectric loss of KNN/PVDF composite films with various volume fractions (3 vol%, 6 vol%, 9 vol%, 12 vol%) is 0.019, 0.022, 0.029 and 0.037 at 1 kHz, respectively. The dielectric loss of KNN-ST/PVDF composite films with different volume fractions (3 vol%, 6 vol%, 9 vol%, 12 vol%) is 0.053, 0.094, 0.201 and 0.419 at 1 kHz, and the detail parameters were shown in Table 1. The dielectric loss of KNN-ST/PVDF composite films was high due to the relaxation polarization loss caused by the interfacial polarization, and the in-homogeneous dispersion of KNN-ST particles aggravates the effect of the interface. It can be seen that the dielectric permittivity of the composite films increases with the increasing concentration of loading KNN and KNN-ST particles. The interfaces between the polymer matrix and particles were the key reason for the variation of dielectric permittivity and loss. The frequency dependent dielectric loss will increase when the dielectric permittivity changes with the frequency, because of the frequency dependent response of the interfacial polarization. In addition, the conductivity increases obviously upon the introduction of SrTiO_3_, which leads to increasing conduction loss. However, the KNN-ST/PVDF composite films possess lower dielectric loss compared with the films loading with BaTiO_3_, PZT and BCZT [30,31,32,33].

The breakdown strength of KNN-ST/PVDF composites films and KNN/PVDF composites films was shown in Figure 6. The results illustrate that KNN-ST/PVDF composite films have higher breakdown strength than KNN/PVDF composite films. This should be attributed to the modification of SrTiO_3_. As is well-known to all, the grain size of filler has an important effect on breakdown strength in the composite materials, and a smaller grain size leading to higher breakdown strength [34,35,36,37,38]. It was found that the breakdown strength of (Ba, Sr)TiO_3_ ceramics increases from 144 to 243 kV/cm with decreasing the grain size from 5.6 to 0.5 μm [39]. The SrTiO_3_ could inhibit grain size growth with the modification of SrTiO_3_ as the microtopographies of the powders. The grain size of KNN-ST particles decreased from 2 μm to 350 nm, which take them much better to distributed homogeneously in PVDF matrix. It could reduce the defects of composite films, and thus improving the breakdown strength of composite films.

In order to investigate the energy storage performance of the composite films, the P-E loops of KNN-ST/PVDF, KNN/PVDF and pure PVDF were measured under the field of 1500 kV/cm at 100 Hz as shown in Figure 7. The polarization of composite films increases with the volume fraction of fillers, which reaches a maximal value 2.88 μC/cm^2^ for 12 vol% KNN-ST/PVDF composite films. It can also be observed that the KNN-ST/PVDF composites films provide a relatively higher saturation polarization and lower remnant polarization. And the decrease of remnant polarization intensity is closely related to the paraelectric behavior of SrTiO_3_. The recoverable energy density of composite films at room temperature is shown in Figure 8. The recoverable energy storage density (W_rec_) of composite films is depended on the concentration of the fillers. The composite films reach the largest W_rec_ with a value of 1.34 J/cm^3^ at 1500 kV/cm for that with 12 vol% KNN-ST concentration, which is twice than that of the KNN/PVDF composite films with the same loading concentration. The results clearly indicate that the KNN-ST fillers of PVDF could significantly improve the discharge energy storage performance [40,41].

For the application of dielectric capacitors in practice, both a high energy storage and a high efficiency (η) are desired simultaneously [42,43,44]. Figure 9 shows the energy storage efficiency of the composite films, the KNN-ST/PVDF composites films show high energy storage efficiency. The maximal efficiency of 74.68% was obtained in KNN-ST/PVDF with a loading of 3 vol% at 1500 kV/cm, which increased 1.29 times than that of the KNN/PVDF composites.

## 4. Conclusions

In summary, the SrTiO_3_ was introduced to reduce the remnant polarization intensity and improve the energy storage density. The KNN and KNN-ST solid solution fillers were fabricated through conventional solid-state reaction route. The PVDF with KNN and KNN-ST fillers with different volume fraction (3 vol%, 6 vol%, 9 vol%, 12 vol%) polymer composites films were prepared by sol-gel method. The effect of KNN-ST modification on the phase structures, microstructures, dielectric properties and energy storage density of PVDF composites films were investigated and discussed in detail. The polymer composites films exhibit an enhanced dielectric constant and improved breakdown strength with the introduction of SrTiO_3_. The optimal dielectric constant of 38 and the maximal discharge energy storage density of 1.34 J/cm^3^ were obtained in KNN-ST/PVDF composites at the concentration of 12 vol% simultaneously. The breakdown strength of composite films was improved upon the introduction of SrTiO_3_, and a breakdown strength 304.44 kV/mm was obtained in 3 vol% KNN-ST/PVDF films. The maximal energy storage efficiency is 74.68% for the 3 vol% KNN-ST/PVDF composite films at 1500 kV/cm. The significant enhancement can be attributed to the paraelectric phase SrTiO_3_. This work provides an accessible route to reduce remnant polarization and improve energy storage performance of the composites.

## Figures and Tables

**Figure 1 polymers-11-00310-f001:**
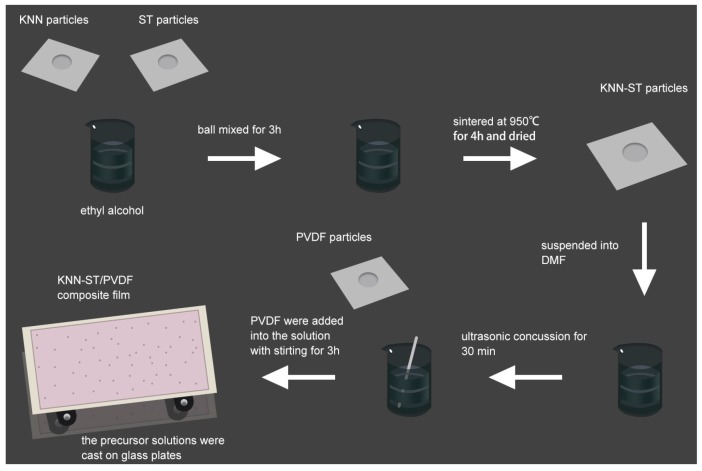
The scheme of the membrane and the particles preparation.

**Figure 2 polymers-11-00310-f002:**
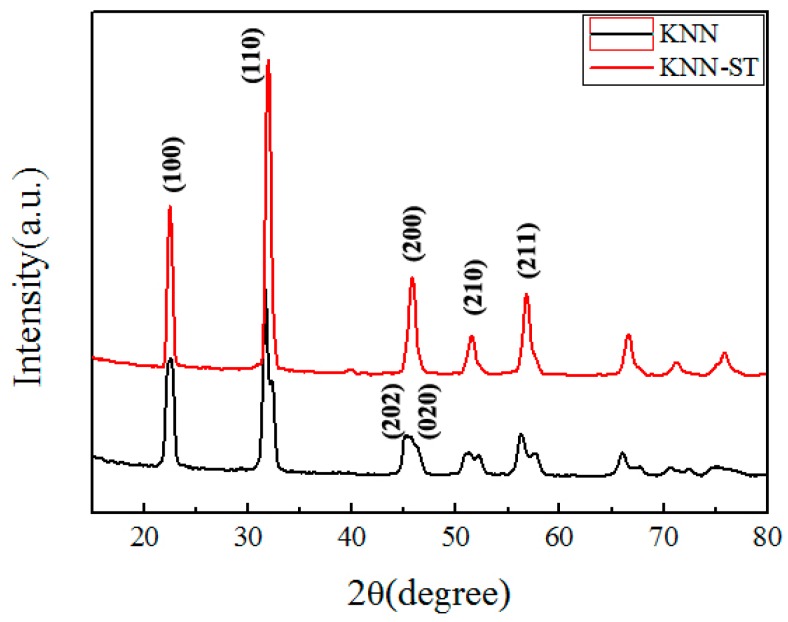
XRD patterns of the K_0.5_Na_0.5_NbO_3_ (KNN) and the (K_0.5_Na_0.5_NbO_3_)-0.15SrTiO_3_ (KNN-ST) particles.

**Figure 3 polymers-11-00310-f003:**
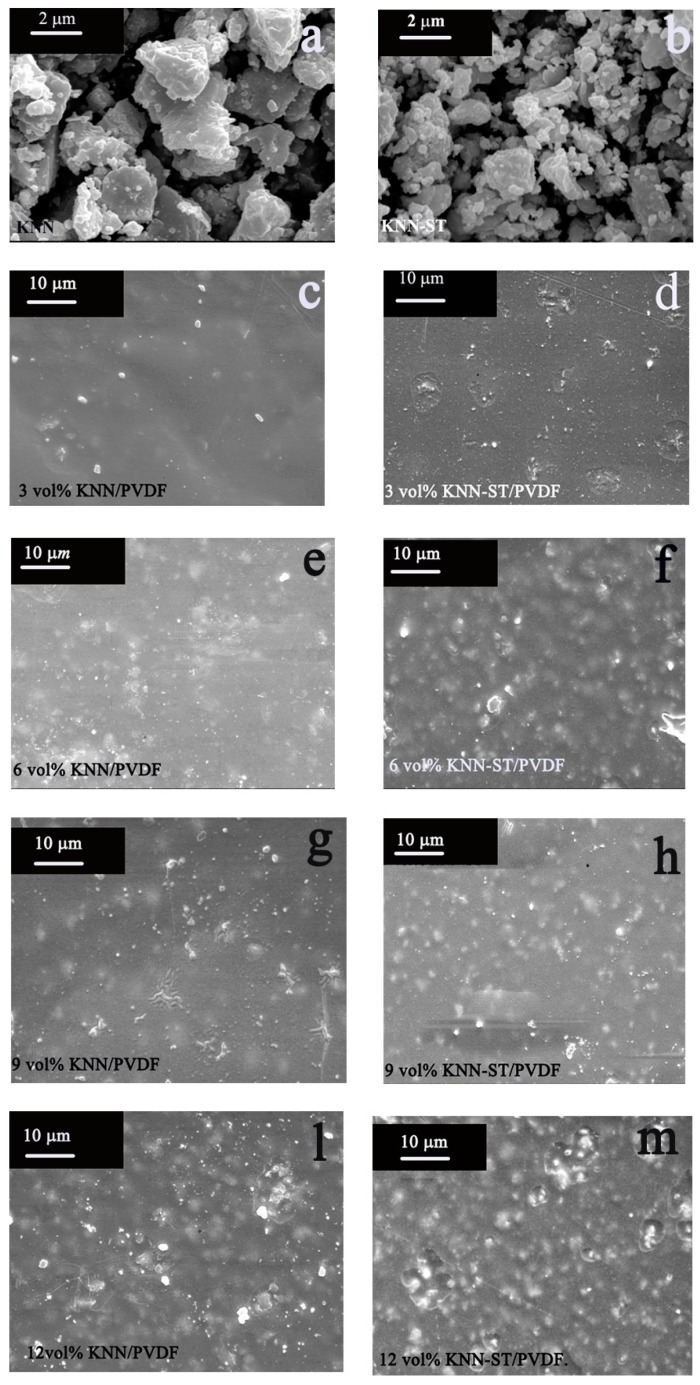
Surface SEM microtopographies of KNN particles, KNN-ST particles, KNN/ polyvinylidene fluoride (PVDF) and KNN-ST/PVDF composite films: (**a**)KNN particles; (**b**) KNN-ST particles; (**c**) 3 vol% KNN/PVDF; (**d**) 3 vol% KNN-ST/PVDF; (**e**) 6 vol% KNN/PVDF; (**f**) 6 vol% KNN-ST/PVDF; (**g**) 9 vol% KNN/PVDF; (**h**) 9 vol% KNN-ST/PVDF; (**l**) 12vol% KNN/PVDF; (**m**) 12 vol% KNN-ST/PVDF.

**Figure 4 polymers-11-00310-f004:**
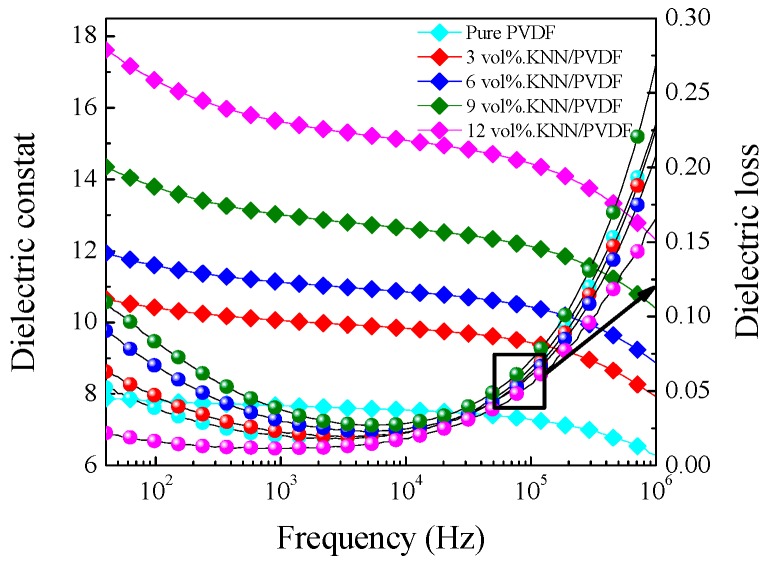
Dielectric constant and dielectric loss on frequency of pure PVDF and KNN/PVDF composite films.

**Figure 5 polymers-11-00310-f005:**
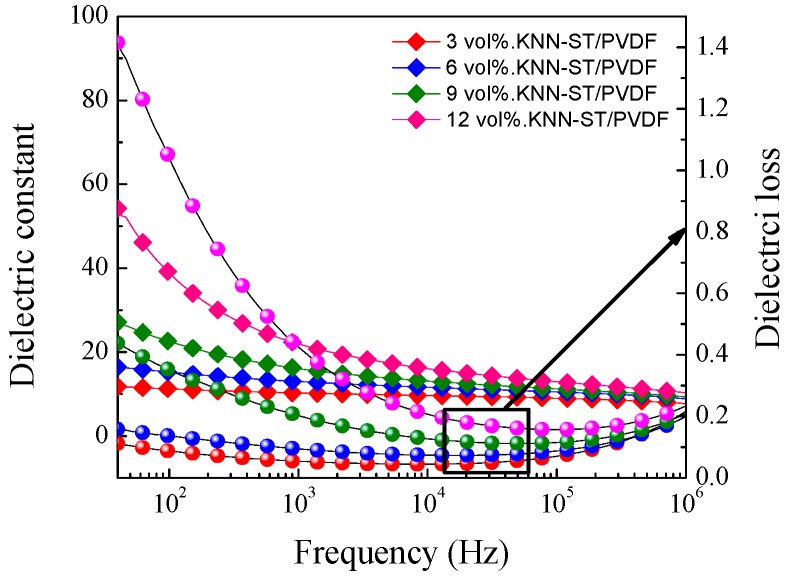
Dielectric constant and dielectric loss on frequency of KNN-ST/PVDF composite films.

**Figure 6 polymers-11-00310-f006:**
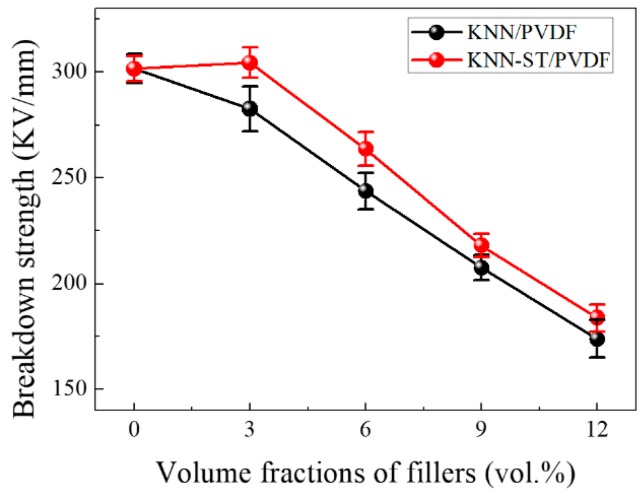
The Breakdown strength of KNN/PVDF and KNN-ST/PVDF composites films.

**Figure 7 polymers-11-00310-f007:**
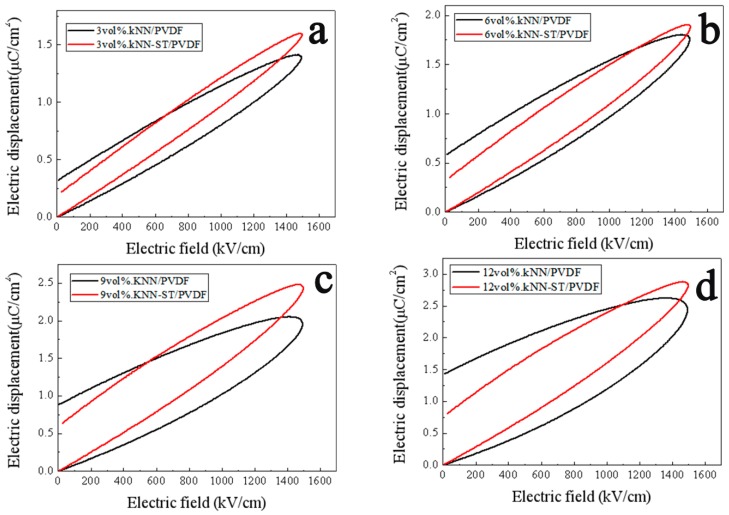
Ferroelectric hysteresis loops of KNN/PVDF and KNN-ST/PVDF composite films.

**Figure 8 polymers-11-00310-f008:**
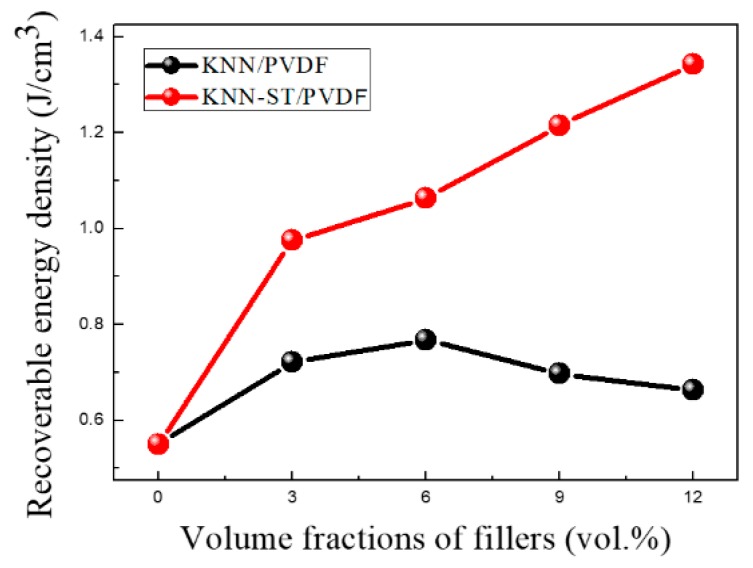
Recoverable energy density of KNN/PVDF and KNN-ST/PVDF composites films.

**Figure 9 polymers-11-00310-f009:**
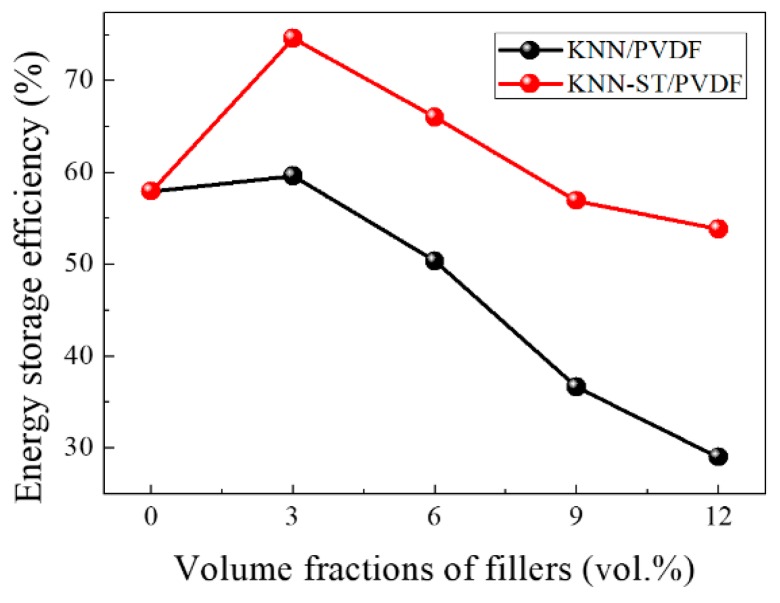
Energy storage efficiency of KNN/PVDF and KNN-ST/PVDF composites films.

**Table 1 polymers-11-00310-t001:** Dielectric properties of pure PVDF, KNN/PVDF and KNN-ST/PVDF composite films.

Pure PVDF	ε_r_	tanδ	P_r_(μC/cm^2^)	W_rec_(J/cm^3^)	η (%)	E_b_(kV/cm)
	8	0.012	0.31	0.55	57.99	301.58
**KNN vol%**	**ε_r_**	**tanδ**	**P_r_(μC/cm^2^)**	**W_rec_(J/cm^3^)**	**η (%)**	**E_b_(kV/cm)**
**3**	10	0.019	0.32	0.72	59.67	282.58
**6**	12	0.022	0.58	0.77	50.34	243.75
**9**	14	0.029	0.88	0.70	36.65	207.57
**12**	17	0.037	1.42	0.66	29.02	173.85
**KNN-ST vol%**	**ε_r_**	**tanδ**	**P_r_(μC/cm^2^)**	**W_rec_(J/cm^3^)**	**η (%)**	**E_b_(kV/cm)**
**3**	11	0.053	0.22	0.98	74.68	304.44
**6**	15	0.094	0.35	1.06	66.02	263.64
**9**	22	0.201	0.64	1.22	56.94	218.01
**12**	38	0.419	0.80	1.34	53.87	183.83

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
