# Peer review of "K0.5Na0.5NbO3-SrTiO3/PVDF Polymer Composite Film with Low Remnant Polarization and High Discharge Energy Storage Density"

_polymers, 2019, doi:10.3390/polym11020310_

Round 1
Reviewer 1 Report
Dear Authors,
The manuscript titled: K0.5Na0.5NbO3-SrTiO3/PVDF Polymer Composite Film 3 with Low Remnant Polarization and High Discharge 4 Energy Storage Density is dealing with the reparation of a composite polymer membrane. The synthesis and characterization procedures are very balanced. I recommend accepting the article after minor revisions. please see my comments below:
1. Abstract 1st sentence please rearrange.
2. please include a table summarising the properties of the prepared membranes.
3. Figure, 3, 4, 5 and 6 can be reduced to a maximum 2 figures and summarising the results in tabulated form.
4. A scheme of the membrane and the ceramic preparation will increase the readability of the article.
Regards,
Author Response
Dear Reviewer 1:
First of all, we acknowledge your comments and suggestions very much, which are valuable in improving the quality of our manuscript. We revised our manuscript in accordance with your instructive guidance. And the revised manuscript is a great improvement on the original manuscript.
Question 1: Abstract 1st sentence please rearrange.
Re: We are grateful for the reviewer’s expertise and insightful comments. According to the comment, we have rearranged the first sentence of the abstract. And the entire abstract was rewritten as below:
“A high recoverable energy storage density polymer composite film has been designed, in which the ferroelectric-paraelectric 0.85(K0.5Na0.5NbO3)-0.15SrTiO3 (abbreviated as KNN-ST) solid solution particles were introduced into polyvinylidene fluoride (PVDF) polymer as functional fillers. The effects of the polarization properties of K0.5Na0.5NbO3 (KNN) and KNN-ST particles on the energy storage performances of KNN-ST/PVDF film were systemically studied. And the introduction of SrTiO3 (ST) was effective in reducing the remnant polarization of the particles, improving the dielectric properties and recoverable energy storage density of the KNN-ST/PVDF films. Compared to KNN/PVDF films, the dielectric permittivity of composite films was enhanced from 17 to 38 upon the introduction of ST. And a recoverable energy storage density of 1.34 J/cm3 was achieved, which is 202.60% lager than that of the KNN/PVDF composite films. The interface between the particles and the polymer matrix was considered to the enhanced dielectric permittivity of the films; and the reduced remnant polarization of the composites was regarded as the improving high recoverable energy storage density. The results demonstrated that combing ferroelectric- paraelectric particles with polymers might be a key method for composites with excellent dielectric permittivity, high energy storage density and energy efficiency.”
Question 2: Please include a table summarising the properties of the prepared membranes.
Re: Thanks for the reviewer’s constructive suggestions. The table summarising the properties of the prepared films was added in the revised manuscript. The related descriptions were also added as below.
Table 1.The dielectric properties of pure PVDF, KNN/PVDF and KNN-ST/PVDF composite films
Pure PVDF | εr | tanδ | Pr(μC/cm2) | Wrec(J/cm3) | η (%) | Eb(kV/cm) |
8 | 0.012 | 0.31 | 0.55 | 57.99 | 301.58 | |
KNN vol% | εr | tanδ | Pr(μC/cm2) | Wrec(J/cm3) | η (%) | Eb(kV/cm) |
3 | 10 | 0.019 | 0.32 | 0.72 | 59.67 | 282.58 |
6 | 12 | 0.022 | 0.58 | 0.77 | 50.34 | 243.75 |
9 | 14 | 0.029 | 0.88 | 0.70 | 36.65 | 207.57 |
12 | 17 | 0.037 | 1.42 | 0.66 | 29.02 | 173.85 |
KNN-ST vol% | εr | tanδ | Pr(μC/cm2) | Wrec(J/cm3) | η (%) | Eb(kV/cm) |
3 | 11 | 0.053 | 0.22 | 0.98 | 74.68 | 304.44 |
6 | 15 | 0.094 | 0.35 | 1.06 | 66.02 | 263.64 |
9 | 22 | 0.201 | 0.64 | 1.22 | 56.94 | 218.01 |
12 | 38 | 0.419 | 0.80 | 1.34 | 53.87 | 183.83 |
Question 3: Figure 3, 4, 5 and 6 can be reduced to a maximum 2 figures and summarising the results in tabulated form.
Re: Thanks for the reviewer’s kind suggestion. Following this advice, we reduced the figures in original pattern. The Figure 3 and 4 in the original pattern were replaced by “Figure R1(Figure 3 in the revised manuscript )”. And Figure 5 and 6 were replaced by “Figure R2”(Figure 4 in the revised manuscript). And all the results were summarized in the Table 1.
Question 4: A scheme of the membrane and the ceramic preparation will increase the readability of the article.
Re: We are grateful for the reviewer’s kind comment. It’s a good idea for the manuscript. In the revised manuscript, a scheme for the membrane and the particles preparation was added in Figure R3(Figure 1 in the revised manuscript ). And we hope the revision would increase the readability of the article.

Reviewer 2 Report
This manuscript shows the effect of SrTiO3 (ST) surface functionalization of
K0.5Na0.5NbO3 (KNN) fillers to be used in PVDF composites for energy storage application. Unfortunately, the manuscript was not well-prepared. The English is not good and the discussion about the mechanisms is vague. I have some comments, which are needed to be addressed carefully:
1. There are so many typos. There are abbreviations without mentioning the main original name. The designations are really vague and sometimes misleading.
2. The abstract should be re-written in a better way. The abstract is not representative of the originality and the novelty of the work, and should be concise and get to the main point.
3. The authors missed the concept of “interfaces” in polymer nanocomposites, which is the dominant parameter in determining the final dielectric and mechanical properties. The authors should provide this concept carefully with explained mechanisms in enhancing dielectric and mechanical properties of polymer nanocomposites in the introduction with aid of these references: Dielectric Polymer Nanocomposites by J. Keith Nelson AND Advanced Materials 2018, 30 (4), 1703624.
4. Section 2.1.; The authors should provide the information regarding the materials, the synthesis and the surface functionalization in the way that the other people can reproduce the experimental methods.
5. Line 83; The “dissolution” is a wrong term, the suspending is right. There are other misused terms such as doping, ceramic and etc. Please check them.
6. Line 100-107; I do not understand the purpose of XRD here, after surface functionalization some crystal plane peaks with Miller indices (hkl) were disappeared. How this confirms the surface functionalization? Did the author can provide another better method such as IR, XPS, Raman,…?
7. Line 113; The grain growth decrease should be explained corresponding to the synthesis with the proper explained mechanism.
8. Line 114-121, Are the SEM from the cross-section? Since in this way it is better to assess the agglomeration and dispersion of the fillers?! Again the authors need to explain the surface coating, dispersion, final dielectric and mechanical properties better as related to the interface concept. The references of IEEE Transactions on Dielectrics and Electrical Insulation 2017, 24 (3), 1396-1404 AND European Polymer Journal 2017, 87, 255-265.
9. Line 151-152, Again the dispersion and interface here is the dominant parameters, which are missing!
10.Line 159-169, Why the authors missed the dielectric properties of pure components such as the fillers and PVDF solely?
Author Response
Response to Reviewer 2 Comments
Dear Reviewer 2:
First of all, we acknowledge your comments and suggestions very much, which are valuable in improving the quality of our manuscript. We revised our manuscript in accordance with your instructive guidance. And the revised manuscript is a great improvement on the original.
Comment 1 : There are so many typos. There are abbreviations without mentioning the main original name. The designations are really vague and sometimes misleading.
Re: Thanks for the reviewer’s insightful comments. We feel sorry our imprecise or vague description makes the reviewer puzzle. Following this advice, the improper spelling and expressions were corrected in detail. Moreover, we have checked the manuscript carefully, and many expressions were revised in the manuscript. We hope the reviewer would find the revised manuscript satisfactory.
Comment 2: The abstract should be re-written in a better way. The abstract is not representative of the originality and the novelty of the work, and should be concise and get to the main point.
Re: Thanks for reviewer’s advice. In order to represent the originality and the novelty of the work, the entire abstract has been rewritten.
“A high recoverable energy storage density polymer composite film has been designed, in which the ferroelectric-paraelectric 0.85(K0.5Na0.5NbO3)-0.15SrTiO3 (abbreviated as KNN-ST) solid solution particles were introduced into polyvinylidene fluoride (PVDF) polymer as functional fillers. The effects of the polarization properties of K0.5Na0.5NbO3 (KNN) and KNN-ST particles on the energy storage performances of KNN-ST/PVDF film were systemically studied. And the introduction of SrTiO3 (ST) was effective in reducing the remnant polarization of the particles, improving the dielectric properties and recoverable energy storage density of the KNN-ST/PVDF films. Compared to KNN/PVDF films, the dielectric permittivity of composite films was enhanced from 17 to 38 upon the introduction of ST. And a recoverable energy storage density of 1.34 J/cm3 was achieved, which is 202.60% lager than that of the KNN/PVDF composite films. The interface between the particles and the polymer matrix was considered to the enhanced dielectric permittivity of the films; and the reduced remnant polarization of the composites was regarded as the improving high recoverable energy storage density. The results demonstrated that combing ferroelectric- paraelectric particles with polymers might be a key method for composites with excellent dielectric permittivity, high energy storage density and energy efficiency.”
Comment 3: The authors missed the concept of “interfaces” in polymer nanocomposites, which is the dominant parameter in determining the final dielectric and mechanical properties. The authors should provide this concept carefully with explained mechanisms in enhancing dielectric and mechanical properties of polymer nanocomposites in the introduction with aid of these references: Dielectric Polymer Nanocomposites by J. Keith Nelson AND Advanced Materials 2018, 30 (4), 1703624.
Re: Thank you for the reviewer’s suggestions.
A.We added the relative concept of "interface" to the revised manuscripts. And the corresponding revised details were shown as below:
The interface between the particles and the polymer matrix was considered to the enhanced dielectric permittivity of the films; and the reduced remnant polarization of the composites was regarded as the improving high recoverable energy storage density. And the interface leads to the local accumulation of space charge and the increase of the macroscopic dipole moment.
B.We have read the references: “Dielectric Polymer Nanocomposites by J. Keith Nelson AND Advanced Materials 2018, 30 (4), 1703624” carefully, and added it into the “References” part. And the corresponding revised details were shown as below:
“This remarkable enhancement in dielectric constant should be attributed to the higher dielectric constant of the ST and the Maxwell-Wagner-Sillars (MWS) interfacial polarization, which is mainly caused by the large difference in the dielectric constant and conductivity between the fillers and the polymer matrix[30].”
Comment 4: Section 2.1.; The authors should provide the information regarding the materials, the synthesis and the surface functionalization in the way that the other people can reproduce the experimental methods.
Re: Thanks for the reviewer’s advice. The detail synthetic processes were shown in the revised Section 2.1.
“2.1. KNN-ST Fillers Preparation
The 0.85(K0.5Na0.5NbO3)-0.15SrTiO3 (abbreviated as KNN-ST) particles were synthesized via conventional solid state reaction route. The K2CO3 (99.0%), Na2CO3 (99.8%), Nb2O5 (99.99%), SrTiO3 (99.8%) and TiO2 (99.99%) were dried and weighed according to stoichiometric ratio as initial powders. These precursor powders were ball mixed 12h by using zirconia balls for well mixture with ethyl alcohol (the quality ratio is 1:1) as grinding media. And the dried powders were sintered at 950℃ for 4h. Finally, the sintered powders were ball mixed another 12h and dried, which was used as inorganic fillers for preparing composite films. The KNN powders were also prepared by solid state reaction method. The sintering process was carried at 850 ℃ for 4h, which is the only difference with KNN-ST particles. ”
Comment 5: Line 83; The “dissolution” is a wrong term, the suspending is right. There are other misused terms such as doping, ceramic and etc. Please check them.
Re: Thanks for reviewer’s suggestions. We checked the words of the whole manuscripts carefully, and the mistake words such as “dissolution”, “doping” and “ceramic” has been revised. The corresponding revised details are highlighted in red, which shown as following.
“Firstly, the KNN-ST powders were suspended into DMF with 30min ultrasonic concussion.”
“Hence, the dielectric loss of KNN/PVDF and KNN-ST/PVDF composite films increased with the increasing of loading content.”
“In order to investigate the microstructure of KNN and KNN-ST particles,a scanning electron microscopy was performed at room temperature.”
Comment 6: Line 100-107; I do not understand the purpose of XRD here, after surface functionalization some crystal plane peaks with Miller indices (hkl) were disappeared. How this confirms the surface functionalization? Did the author can provide another better method such as IR, XPS, Raman,…?
Re: Thanks for the reviewer’s constructive suggestions. In order to present the structure of KNN and KNN-ST particles, the Miller indices (hkl) in the XRD patterns were added in the revised manuscript. The KNN and KNN-ST particles possess a perovskite structure in this work. The KNN-ST particles were not surface functionalization ones of KNN particles, which were solid solution of KNN and ST. In other word, the ST was introduced in the crystal structure of KNN, the KNN-ST powders were the solid solution particles rather than the surface functionalization of ST to KNN particles. And the revised XRD patterns were shown as following(Figure 1).
Figure R1.XRD patterns of KNN and KNN-ST particles.
Comment 7: Line 113; The grain growth decrease should be explained corresponding to the synthesis with the proper explained mechanism.
Re: Thanks for the reviewer’s kind advice. The KNN-ST particles were solid solution of KNN and ST in this work.The perovskite structure of the particles were synthesized in the solid state reaction of the raw materials. And the grain sizes of KNN and KNN-ST were grown in the high temperature process. Compared with KNN, the crystal structure of KNN-ST was disordered by the introduction of ST. The disordered crystal structure and the growth energy restricted the grain growth of KNN-ST particles.
Figure R2.The scheme of the membrane and the particles preparation
Comment 8: Line 114-121, Are the SEM from the cross-section? Since in this way it is better to assess the agglomeration and dispersion of the fillers?! Again the authors need to explain the surface coating, dispersion, final dielectric and mechanical properties better as related to the interface concept. The references of IEEE Transactions on Dielectrics and Electrical Insulation 2017, 24 (3), 1396-1404 AND European Polymer Journal 2017, 87, 255-265.
Re: Thanks for the reviewer’s suggestions.
A. The SEM is a surface scanning image, which is one of the means to characterize the microstructure of samples. The KNN-ST powders were the solid solution particles, and the fillers are only dispersed into the polymer matrix without surface modification.
B. The references: “IEEE Transactions on Dielectrics and Electrical Insulation 2017, 24 (3), 1396-1404 AND European Polymer Journal 2017, 87, 255-265” was added in the revised manuscript(reference 26).
Comment 9: Line 151-152, Again the dispersion and interface here is the dominant parameters, which are missing!
Re: We highly appreciate the reviewer’s comment. In the revised manuscript,the effects of interface and dispersion on dielectric performance of composite films were discussed as as follow :
“The dielectric loss of KNN-ST/PVDF composite films was high due to the relaxation polarization loss caused by the interfacial polarization, and the inhomogeneous dispersion of KNN-ST particles aggravates the effect of the interface. It can be seen that the dielectric permittivity of the composite films increase with the increasing concentration of loading KNN and KNN-ST particles. The interfaces between the polymer matrix and particles were the key reason for the variation of dielectric permittivity and loss. The frequency dependent dielectriclosswill increase when the dielectric permittivity changes with the frequency, because of the frequency dependent response of the interfacial polarization.In addition, the conductivity increases obviously upon the introduction of SrTiO3, which leads to the increase of conduction loss.”
Comment 10: Line 159-169, Why the authors missed the dielectric properties of pure components such as the fillers and PVDF solely?
Re: In accordance with reviewer’s instructive guidance, the dielectric properties of all samples (including pure PVDF) were provided in Table 1.
Table 1.The dielectric properties of pure PVDF, KNN/PVDF and KNN-ST/PVDF composite films
Pure PVDF | εr | tanδ | Pr(μC/cm2) | Wrec(J/cm3) | η (%) | Eb(kV/cm) |
8 | 0.012 | 0.31 | 0.55 | 57.99 | 301.58 | |
KNN vol% | εr | tanδ | Pr(μC/cm2) | Wrec(J/cm3) | η (%) | Eb(kV/cm) |
3 | 10 | 0.019 | 0.32 | 0.72 | 59.67 | 282.58 |
6 | 12 | 0.022 | 0.58 | 0.77 | 50.34 | 243.75 |
9 | 14 | 0.029 | 0.88 | 0.70 | 36.65 | 207.57 |
12 | 17 | 0.037 | 1.42 | 0.66 | 29.02 | 173.85 |
KNN-ST vol% | εr | tanδ | Pr(μC/cm2) | Wrec(J/cm3) | η (%) | Eb(kV/cm) |
3 | 11 | 0.053 | 0.22 | 0.98 | 74.68 | 304.44 |
6 | 15 | 0.094 | 0.35 | 1.06 | 66.02 | 263.64 |
9 | 22 | 0.201 | 0.64 | 1.22 | 56.94 | 218.01 |
12 | 38 | 0.419 | 0.80 | 1.34 | 53.87 | 183.83 |

Reviewer 3 Report
I believe that the paper entitled "K0.5Na0.5NbO3-SrTiO3/PVDF Polymer Composite Film with Low Remnant Polarization and High Discharge Energy Storage Density" is suitable for publication in Polymers. I suggest to check the following points prior publications:
You need to replace "combing" with combining" along your manuscript.
I suggest to have a Table with the polymer composite films and their performance in Introduction. It is not easy to follow the information in the text.
You need to mention in 2.2 KNN-ST/PVDF Polymer Composites Films Preparation how you prepared the composite films; doping levels etc.
Please pay attention in English grammar of the following sentence
"..., there is no impurity phase could be found."
The following sentence has no verb
"The breakdown strength of KNN-ST/PVDF composites films and KNN/PVDF composites films as shown in Figure 7."
Replace "is dependents" with "is depended".
Replace "were obtain" with "were obtained".
Author Response
Response to Reviewer 3 Comments
Dear Reviewer 3:
First of all, we acknowledge your comments and suggestions very much, which are valuable in improving the quality of our manuscript. We revised our manuscript in accordance with your instructive guidance. And the revised manuscript is a great improvement on the original.
Comment 1: You need to replace "combing" with "combining" along your manuscript.
Re: Thanks for the reviewer’s advice. The common mistake word in the text has been replaced.The corresponding revised details are highlighted and can be found as follow:
“The results demonstrated that the high dielectric constant ceramic combining with the para-electric hybrid fillers for composites exhibit low remnant polarization, high energy storage density and high energy efficiency.”
“In order to improve the dielectric constant, an effective strategy that combining the polymer composites with the ceramic fillers has been recently carried out, and the electrical, optical or magnetic performance can be tailored by adjusting the shape and size and modifying the surface of the filled particles.”
Comment 2: I suggest to have a Table with the polymer composite films and their performance in Introduction. It is not easy to follow the information in the text.
Re: The table summarising the properties of the prepared membranes was added in the revised manuscript. The corresponding revised details are highlighted in red and can be found in Table 1 of the revised manuscript. And we hope you will find the revised manuscript satisfactory.
Table 1.The dielectric properties of pure PVDF, KNN/PVDF and KNN-ST/PVDF composite films
Pure PVDF | εr | tanδ | Pr(μC/cm2) | Wrec(J/cm3) | η (%) | Eb(kV/cm) |
8 | 0.012 | 0.31 | 0.55 | 57.99 | 301.58 | |
KNN vol% | εr | tanδ | Pr(μC/cm2) | Wrec(J/cm3) | η (%) | Eb(kV/cm) |
3 | 10 | 0.019 | 0.32 | 0.72 | 59.67 | 282.58 |
6 | 12 | 0.022 | 0.58 | 0.77 | 50.34 | 243.75 |
9 | 14 | 0.029 | 0.88 | 0.70 | 36.65 | 207.57 |
12 | 17 | 0.037 | 1.42 | 0.66 | 29.02 | 173.85 |
KNN-ST vol% | εr | tanδ | Pr(μC/cm2) | Wrec(J/cm3) | η (%) | Eb(kV/cm) |
3 | 11 | 0.053 | 0.22 | 0.98 | 74.68 | 304.44 |
6 | 15 | 0.094 | 0.35 | 1.06 | 66.02 | 263.64 |
9 | 22 | 0.201 | 0.64 | 1.22 | 56.94 | 218.01 |
12 | 38 | 0.419 | 0.80 | 1.34 | 53.87 | 183.83 |
Comment 3: You need to mention in 2.2 KNN-ST/PVDF Polymer Composite Films Preparation how you prepared the composite films; doping levels etc.
Re: Thanks for the reviewer’s kind suggestion. According to the comment, we have added the loading levels of the polymer composite films. And the detail preparation process of KNN-ST/PVDF polymer composite films are highlighted in red and can be found in 2.2 .
“2.2. KNN-ST/PVDF Polymer Composites Films Preparation
The sol-gel method was taken to prepare the KNN/PVDF and KNN-ST/PVDF polymer composites films (with 3vol%, 6vol%, 9vol% and 12vol% loading concentration respectively). The KNN-ST particles and PVDF powders were used as raw materials and the dimethylformamide (DMF) were used as the solvent. Firstly, the KNN-ST powders were suspended into DMF with 30 min ultrasonic concussion. Then, the PVDF particles were added into the solution with stirring for 3h. In order to prevent the residual bubbles in the solutions, the precursor solutions were placed into a vacuum drying oven for 2h. And the precursor solutions were cast by an automatic film applicator. The obtained polymer composite films were heated at 100℃ for 24h to evaporate the solution. Finally, press the composite films with a vulcanizing press. The thickness of the films ranges from 20 to 40 μm.”
Comment 4: Please pay attention in English grammar of the following sentence"..., there is no impurity phase could be found." The following sentence has no verb "The breakdown strength of KNN-ST/PVDF composites films and KNN/PVDF composites films as shown in Figure 6." Replace "is dependents" with "is depended". Replace "were obtain" with "were obtained".
Re:Thanks for the reviewer’s insightful comments. The improper sentences have been corrected or deleted, and the mistake word in the manuscript has been replaced.
A.The sentence "..., there is no impurity phase could be found." has been deleted.
B. Other errors in the text have been corrected as below.
The breakdown strength of KNN-ST/PVDF composites films and KNN/PVDF composites filmswas shown in Figure 6.
The recoverable energy storage density (Wrec) of composite films is depended on the concentration of the fillers.
The optimal dielectric constant of 38 and the maximal discharge energy storage density of 1.34 J/cm3 were obtained in KNN-ST/PVDF composites at the concentration of 12 vol% simultaneously.

Round 2
Reviewer 2 Report
The authors provided fair enough discussion according to my comments. There are missing information:
Previous comments 3 and 8: There are missing two references,Please check carefully. These two references are:
(1) Dielectric Polymer Nanocomposites by J. Keith Nelson
(2) EEE Transactions on Dielectrics and Electrical Insulation 2017, 24 (3), 1396-1404
Previous comment 6: Again the discussion is vague, if two crystal structure doesn't make any superlattice. It should be two different crystal structure without overlapping the XRD peaks.